# Combined Method Comprising Low Burden Physiological Measurements with Dry Electrodes and Machine Learning for Classification of Visually Induced Motion Sickness in Remote-Controlled Excavator

**DOI:** 10.3390/s24196465

**Published:** 2024-10-07

**Authors:** Naohito Yoshioka, Hiroki Takeuchi, Yuzhuo Shu, Taro Okamatsu, Nobuyuki Araki, Yoshiyuki Kamakura, Mieko Ohsuga

**Affiliations:** 1Graduate School of Robotics and Design, Osaka Institute of Technology, Chayamachi 1-45, Osaka 530-0013, Japan; hiroki.oit@gmail.com; 2Research and Development Center, Yanmar Holdings Co., Ltd., Umegahara 2481, Maibara 521-8511, Japantaro_okamatsu@yanmar.com (T.O.); nobuyuki_araki@yanmar.com (N.A.); 3Faculty of Information Science and Technology, Osaka Institute of Technology, Kitayama 1-79-1, Hirakata 573-0196, Japan; yoshiyuki.kamakura@oit.ac.jp; 4Faculty of Robotics and Design, Osaka Institute of Technology, Chayamachi 1-45, Osaka 530-0013, Japan; mieko.ohsuga@oit.ac.jp

**Keywords:** visually induced motion sickness, cybersickness, physiological measurement, machine learning, remote control, low-burden physiological measurement, operator condition

## Abstract

The construction industry is actively developing remote-controlled excavators to address labor shortages and improve work safety. However, visually induced motion sickness (VIMS) remains a concern in the remote operation of construction machinery. To predict the occurrence and severity of VIMS, we developed a prototype system that acquires multiple physiological signals with different mechanisms under a low burden and detects VIMS from the collected data. Signals during VIMS were recorded from nine healthy adult males operating excavator simulators equipped with multiple displays and a head-mounted display. Light gradient-boosting machine-based VIMS detection binary classification models were constructed using approximately 30,000 s of time-series data, comprising 23 features derived from the physiological signals. These models were validated using leave-one-out cross-validation on seven participants who experienced severe VIMS and evaluated through area under the curve (AUC) scores. The mean receiver operating characteristic curve AUC score was 0.84, and the mean precision–recall curve AUC score was 0.71. All features were incorporated into the models, with saccade frequency and skin conductance response identified as particularly important. These trends aligned with subjective assessments of VIMS severity. This study contributes to advancing the use of remote-controlled machinery by addressing a critical challenge to operator performance and safety.

## 1. Introduction

In recent years, the construction industry in Japan has been actively using information and communication technology to improve productivity and safety and address labor shortages [1]. The remote operation of construction machinery has emerged as a solution that is expected to improve the working environment by eliminating workers’ need to visit construction sites. Operating the machine remotely from an office eliminates noise and vibration while ensuring the safety of the operators. However, there are operability challenges, such as difficulty in assessing the surrounding environment and accurately perceiving depth as the operation is performed through a display. Moreover, health problems, such as visually induced motion sickness (VIMS) [2,3], can come from prolonged viewing of a display while performing tasks. Several studies have proposed solutions to the operability issues [4]. However, few studies have been conducted on the health of the operators. Therefore, we focused on VIMS during remote-controlled operations and investigated methods to prevent it.

VIMS, a type of motion sickness characterized by symptoms such as dizziness, nausea, and discomfort, typically occurs when an image presented in a wide field of view (FOV) rotates around the operator. The work on an excavator is consistent with this condition because it involves the rotation of the cabin during operation [5,6]. Continuing work after the onset of VIMS is not recommended; operators should be given adequate rest and medical care. Once VIMS develops, recovery takes time. Moreover, symptoms may develop without the operator’s awareness. Some operators may attempt to finish their tasks despite experiencing symptoms, leading to a worsening of their condition. Therefore, detecting early signs of VIMS in operators and preventing them from working once these signs are confirmed is essential. This study aimed to develop such an early warning system.

## 2. System Overview

VIMS affects the autonomic nervous system [3] and brain activity [7,8]. Therefore, we have prototyped a system [9] that detects VIMS based on physiological signals related to the autonomic nervous system and the electroencephalogram (EEG). The configuration of the measurement system is shown in Figure 1.

The system comprises two physiological signal-recording devices with dry electrodes (Figure 2) [10,11,12] and sensors. In conventional methods, electrodes are attached to the skin using conductive paste and tape to measure the electrocardiogram (ECG), skin electrodermal activity (EDA), electrooculogram (EOG), and EEG. This requires time and effort to fix the electrodes; the use of dry electrodes mitigates this problem. Electrodes and sensors are attached to levers (Figure 3a), seatbelts (Figure 3b), helmets, or head-mounted displays (Figure 3c). The prototype system can acquire signals from the ECG, photoplethysmogram (PPG), EDA, respiration (Resp), the horizontal component of the EOG, and EEG from four locations (T5, P3, P4, and T6) based on the International 10-20 system [13]. Thus, the proposed system requires no devices other than those essential for normal operation, and the physiological signals can be measured as soon as the operator is seated in the cockpit and grasps the lever. A comparison of the system with conventional methods confirmed that this system provides reasonably accurate measurements [9]. Although the raw signals obtained by the system contained noise, they could be used as indicators through frequency filtering, signal period conversion, and averaging.

## 3. Acquisition of Data

The data were obtained in accordance with the report of the Life Science Ethics Committee of the Osaka Institute of Technology (No. 2022-19-1). The participants were healthy, paid adults who provided written informed consent. A total of 11 males aged between 24 and 61 years (average age: 45.7 ± 9.0 years) with experience in operating a hydraulic excavator participated.

### 3.1. Experimental Setting

The experiment was conducted in an environment that simulated the actual operation of a remote-controlled hydraulic excavator (Figure 4). Two different video display devices, a multi-display (MD) and a head-mounted display (HMD), were used in the experiment. The experiment was conducted on different days with two conditions per participant. The order of the two conditions was counterbalanced among the participants. The experimental scenario was created using computer graphics and an operating simulator (Vortex Studio Create, CM Labs Simulations). This was a simulated excavation task [14] that involved lifting an oil drum, turning it by 180°, placing it on the ground, and repeatedly returning it to its original position (Figure 5).

The participants verbally reported their VIMS subjective ratings, drowsiness, and fatigue on a five-point scale each time the drum was placed on the ground. The VIMS subjective ratings were as follows: 1 = not uncomfortable at all (not aware of VIMS); 2 = slightly uncomfortable (slightly aware of VIMS); 3 = uncomfortable (aware of VIMS); 4 = very uncomfortable (operating while tolerating VIMS); and 5 = extremely uncomfortable (wanting to quit immediately) [15]. The fast-motion sickness scale (FMS) [16] is often used as a subjective rating tool for VIMS. However, in this study, unlike previous research, VIMS ratings were collected during excavator operation rather than during video viewing. To minimize task interruption caused by hesitation when using a detailed scale, such as the FMS, and to account for the simultaneous reporting of other ratings, we used a five-point scale with specific descriptions of conditions for the VIMS rating. Additionally, the primary goal of our system is to binarily classify whether the operator is sufficiently fit to work. The experiment was terminated when the degree of VIMS reached its maximum value (5), the operation time reached 1 h, or the restraint time reached 2 h. In some cases, two hours were required to respond to unintended troubles such as operation simulator malfunctions and poor Bluetooth transmission. The participants also completed a simulator sickness questionnaire (SSQ) [17] and a questionnaire on their feelings of fatigue [18] before and after the experiment. In addition to physiological signals, facial images from a camera (only in the MD session) [19], head speed, angular velocity, and excavator operation logs were recorded (Figure 1).

### 3.2. Effect of Inducing VIMS

Two of the 11 participants were unable to record their physiological signals because of Bluetooth communication problems with the system; thus, valid data were available for nine participants.

Figure 6 presents the average SSQ scores of the nine participants. The SSQ scores significantly increased after the experiment, indicating that VIMS was induced. Table 1 presents the VIMS values reported for subjective ratings. Of the nine participants, six (P02-07) were unable to complete the excavation task because of severe VIMS. Three of them (P02-04) experienced severe VIMS under both display conditions, whereas the others (P05-07) experienced VIMS only with the head-mounted display. For one of the nine participants (P01), the VIMS level increased; however, the operator continued with the task while enduring VIMS symptoms. Two participants (P08 and P09) continued with the operation with only slight VIMS. The mean experimental times for all participants were 1616 s for the MD and 1501 s for the HMD.

## 4. Model Construction

We constructed a machine learning model to detect VIMS using a light gradient-boosting machine (LightGBM) [20,21,22], which is an algorithm that uses gradient boosting. Among the gradient-boosting algorithms, LightGBM is very fast in data processing and maintains high accuracy in classification performance. In addition, it can visualize the importance of the features in the model.

A binary classifier was constructed by designating a VIMS level of 3 or higher as the level at which the operator was not recommended to continue the task owing to the severity of VIMS. Twenty-three features obtained from physiological signals were used as explanatory variables in the model.

### 4.1. Preprocessing Method

The VIMS level for the nine participants and the physiological signals, except for PPG, were processed into data for constructing the machine learning model; PPG was not used in this study because it has numerous missing values, poor measurement accuracy, and is similar to ECG. All physiological signals were converted into a time series of machine learning data indices (features) at 1-second intervals. The values at that time were the representative indices obtained from the latest 30-second window, shifting by 1 s. The time window was set so that physiological indices could be calculated with the minimum time resolution. The 30-second window was established as the minimum duration required to compute the frequency component of heart rate (HR) [23].

#### 4.1.1. VIMS Level

The VIMS levels were converted to time-series data by updating the values each time (report in Figure 5); a participant reported a subjective rating. The maximum score was 4 because the operation was stopped when the level reached 5. These data were also used as objective variables after conversion to binary values of “1 or 2” and “3 or 4”.

#### 4.1.2. ECG

After band-pass filtering (0.05–30 Hz), the RR interval was calculated by R-wave enhancement filtering [24] and converted to the instantaneous HR. The instantaneous HR was a third-order spline interpolated and resampled at 50 Hz at equal intervals. The frequency components of the HR were analyzed, and the low-frequency component (LF) and high-frequency component (HF) were calculated from the spectral components from 0.04–0.15 Hz and 0.15–0.40 Hz. The mean HR, standard deviation of HR, LF, HF, and LF/HF for a 30-second window were used as features.

#### 4.1.3. Resp

The Resp period was calculated by applying a band-pass filter (0.5–2 Hz) to the accelerometer signal and a difference filter [25]. The mean and standard deviation of the Resp period for a 30-second window were used as features.

#### 4.1.4. EDA

The skin conductance signal measured by the EDA was low-pass filtered at a cutoff frequency of 0.04 Hz to obtain the skin conductance level (SCL). The mean value and standard deviation of the SCL for 30 s were used as the features. In addition, noise components were extracted by applying a high-pass filter with a cutoff frequency of 0.5 Hz to the original signal. The skin conductance response (SCR) was obtained by subtracting the SCL from the output after removing the noise component from the original signal. The integral of the absolute value of the SCR in a 30-second window was used as a feature.

#### 4.1.5. EOG

The number of horizontal saccade eye movements per second was calculated from the 30-second signal. The peak value of the time derivative of the potential change in the EOG of each saccade (the value corresponding to the saccade speed [26,27]) was extracted. The saccade frequency and mean and standard deviation of the saccade speed for 30 s were used as the features.

#### 4.1.6. EEG

After band-pass filtering (0.5–30 Hz) and averaging of the 4-channel EEG signals, the frequency components were analyzed. The power ratios and peak frequencies of alpha-component (8–13 Hz) and theta-component (4–8 Hz) were calculated for a 2-second time window. The mean and standard deviation of the power ratio and peak frequency for 30 s were used as features.

In addition, the eye-fixation-related potential (EFRP) [28] was calculated by averaging the EEG signals for each eye fixation. The EFRPs of the four channels were averaged. The amplitude of the lambda response was defined as the maximum value between 50 and 200 ms, with each eye fixed at 0 ms along the time axis. The mean and standard deviation of the amplitude and latency of the lambda response for 30 s were used as features.

#### 4.1.7. Standardization and Interpolation of Missing Values

Each feature was standardized by converting it to a ratio of the reference value for each experiment, using the mean value of 90 data points during the first 120 s of the experiment as the reference value. In general, it is expected that model accuracy can be improved by converting each feature to a *Z*-score for all data during preprocessing for standardization. However, in this case, the physiological signals can be adapted to a real-time system by standardizing the data for a few seconds immediately after the start of a steady state.

After converting the data into ratios, the data for all participants were combined for each feature, and histograms were created. The 1 and 99 percentile values were used as the lower and upper thresholds, respectively. Values below or above these thresholds were considered abnormal. In the future, these thresholds can be adapted for real-time systems. The error values are interpolated by reusing the values from the previous timeframe.

### 4.2. Parameter Tuning and Validation Method

The number of time-series segments included in each dataset obtained from the nine participants is shown in Table 2; this is divided into binary values of VIMS severity. The model was evaluated using leave-one-out cross-validation (LOOCV) on seven participants (P01-07) who developed severe VIMS (VIMS level 3 or 4) with at least one condition (FD or HMD). In LOOCV, the data excluding one participant’s data were used as the training data for parameter tuning and training. The data removed in advance were used as the test data to verify the quality of the model. Thus, the generalizability to unknown data was guaranteed.

The hyperparameters were tuned using a combination method comprising the Optuna module [29,30,31] and stratified 5-fold cross-validation using the training data. The LightGBMTuner [28] in the Optuna module automatically searched for the best combination of the seven parameters (lambda_l1, lambda_l2, num_leaves, feature_fraction, bagging_fraction, bagging_freq, and min_child_samples). The LightGBMTunner was a library specialized for tuning the hyperparameters of the LightGBM. During the parameter tuning, class weights were applied based on the amount of data in each class.

### 4.3. Evaluation Index for Classification Model

The accuracy, the Kappa coefficient [32], and the area under the curve (AUC) were used to evaluate the model. Accuracy is a measure of how well the overall prediction result matches the true value. The Kappa coefficient is an index used in the field of medicine to evaluate the agreement between the diagnoses of two observers. In the field of machine learning, the test data can be regarded as one observer, the prediction data of the model as the other, and the degree of agreement between them can be evaluated. Other indicators can be used to evaluate the model using the true positive rate (TPR or Recall), false positive rate (FPR), and precision. These values change with the threshold for separating the binary values. Generally, the receiver operating characteristic curve AUC (ROC-AUC) score is used as an index of the quality of binary classifiers. However, this score may be overestimated when there is a large bias in the number of samples among classes; some of the test datasets (P05 and P07) have a bias, as shown in Table 2. Therefore, both the ROC-AUC and precision–recall curve AUC (PR-AUC) were used as indices to evaluate model quality.

## 5. Results

As shown in Table 3, the mean accuracy was 0.81 with LOOCV, and the maximum accuracy was 0.91, with all but one participant scoring > 0.7 (Table 3). The mean Kappa coefficient was 0.45, with a maximum value of 0.73. The upper scores were reasonably good; however, the lower values can be improved. The mean ROC-AUC score was 0.84. The maximum ROC-AUC score was 0.96, with all participants scoring > 0.7. The mean PR-AUC score was 0.71, with a maximum value of 0.98. For P05 and P07, where the number of classes was biased, the score of P05 was 0.59, which was adequate, but in the case of P07, the PR-AUC score was only 0.38. In the case of P07 (Figure 7g), the recall was only 0.09, with a precision rating of 0.5.

The feature importance of each model and the average for all models are shown in Figure 8. The most important features in these models were those related to the EOG and EDA, which were saccade frequency and SCR.

## 6. Discussion

The mean accuracy and Kappa coefficient in this study with LOOCV were better than the accuracy of 0.66 and the Kappa coefficient of 0.28 realized in a previous study by Liu et al. [8]. In addition, the mean AUC score in this study with LOOCV was better than the AUC score of 0.75 in a previous study by Keshavarz et al. [3] and 0.68 by Liu et al. [8]. The model in the study by Keshavarz et al. was constructed primarily using autonomic nervous system indices, whereas the model in the study by Liu et al. was based on EEG. The model used in this study utilizes a variety of physiological indices. Numerous features related to the physiological responses with different mechanisms measured in this study were used as important features (Figure 8). The quality of the model appears to have improved because the inability to evaluate the VIMS level with one feature can be compensated by another feature.

To consider model improvement and to clarify the relationship between VIMS and physiological indices, trends in the accuracy rate, the VIMS level, and changes in each feature were identified from the data in which VIMS was severely affected. As the time transition of VIMS differed for each participant in the experiment, the time length of the data also differed. Therefore, the time-series data were resampled into 1000 segments with a start time of 0 and an end time of 1 to create a standardized time axis, and the data were averaged for each segment. For the accuracy rate, the time axis of the data with true positives and true negatives set to 1 and false positives and false negatives set to 0 were standardized. Figure 9 shows the mean change in the accuracy rate, the VIMS level, and each feature. The mean VIMS levels increased with time (Figure 9b). The 95% confidence interval around VIMS levels 2–3 was wide because VIMS may gradually or rapidly become more severe.

### 6.1. Accuracy Rate

The accuracy rate was high during the healthy period soon after the operation started and the period when the VIMS became severe, just before the operation was stopped (Figure 9a). However, the value was lower around VIMS levels 2–3 in the middle of the entire period. There were several reasons for this observation. One reason was that the participants were unable to accurately evaluate the degree of VIMS. They do not often experience VIMS in their daily lives, such as while watching movies or playing games. Another reason is that the timing of the change in the VIMS level and the timing of the subjective rating report were not synchronized. In this study, no participant experienced a decrease in VIMS levels during the operation. Therefore, only the time when the VIMS score changed from 2 to 3 was considered affected by the timing discrepancy. Consequently, the effect on the accuracy of the classification is considered very small because the period during which the deviation occurs is only a small part of the total time period. Finally, the effects of unstable physiological responses were considered. When there is an adverse change in physical condition, the normal physiological response and the response in a bad physical condition are repeated to maintain normal physiological conditions. To solve the problems of subjective evaluation and unstable physiological responses, it is necessary to consider a method for evaluating operators by observing the changes in their work performance. Another approach would be to construct a second model specialized for classification at VIMS level 2 or 3 and refer to the outputs of the second model during the time range in which classification in the first model is not stable.

### 6.2. ECG

The HR increased slightly with time (Figure 9c), indicating that the HR increased with the severity of VIMS, as described in previous studies [3,33,34,35]. This slight change may have contributed to the model. The pattern of change was a gradual increase up to a moderate level of VIMS, followed by repeated increases and decreases to a level slightly higher than the steady state. This trend was due to the sympathetic nervous system being aroused by the stress caused by VIMS.

### 6.3. Resp

As time progressed, the Resp period was prolonged (Figure 9d); that is, the Resp rate (RR) decreased. This indicates that the RR decreased with the severity of VIMS. The changing trend was almost constant up to moderate levels of VIMS and was more pronounced at the level of severe VIMS. Previous studies have reported that RR increases with VIMS severity [35,36] or, conversely, has a negative correlation [37,38,39]. The former is a case of moderate severity of VIMS, and the effects of cognitive load, anxiety about the sickness, and time course are also suspected. In the latter case, breath-holding may have occurred as a defensive reaction to vomiting caused by severe VIMS [40]. In addition, transient slow large respiration is observed with moderate car or simulator sickness, which causes an increase in the Resp center-of-gravity frequency and instability [15]. This response, whether intentional or unintentional, is thought to be a defensive reaction in which the human body attempts to return to a normal state. In the present study, respiratory rate fluctuations also increased over time (Figure 9e), suggesting that this phenomenon occurred; however, further studies, including close examination of waveforms, are needed to clarify the relationship between respiration and VIMS.

### 6.4. EDA

Both SCL_mean and SCR increased over time (Figure 9f,g), indicating that they increased with the VIMS severity, as described in previous studies [41]. SCL_mean increased soon after the start of the experiment and then remained higher than the baseline. The SCR increased gradually after the start of the experiment and further increased with increasing VIMS severity [42]. This trend is similar to that of the change in VIMS level, which may have been one of the most important features of the model.

### 6.5. EOG

Over time, the saccade frequency decreased, and the saccade speed increased (Figure 9h,i). These results were consistent with those of previous studies [43,44]. The decrease in saccade frequency shows a trend very similar to that of the change in VIMS level, suggesting that this is the most important feature. Since it has been suggested that VIMS may be related to optical flow in the retina, the decrease in eye movements may have occurred as a defense response to control the progression of VIMS. The saccade speed increased at a moderate VIMS level and then remained above the baseline. Wibirama et al. reported that the eye movement speed increased in an environment that also induced severe VIMS [43]. In the previous study, it was considered that the increase in eye movement speed was attributed to the optical flow [44], and the high speed of the optical flow caused severe VIMS. In this study, the scene with the largest change in optical flow was the scene of cabin rotation during operation (Figure 5). The cabin rotation speed did not change from the beginning to the end of the experiment. Therefore, the change in saccade speed in this study was not induced by the optical flow. It is thought that the increase in the saccade speed as a defensive response to VIMS may have prevented the perception of the change in the optical flow on the retina. To verify this, further studies focusing on eye movements are needed.

### 6.6. EEG

Regarding the frequency components of the EEG, the theta power ratio increased, and the alpha power ratio decreased with time (Figure 9j,k). The peak alpha frequency shifted to the lower frequency side (Figure 9m), indicating an increase in the proportion of the theta power ratio. These results are consistent with those of previous studies [38,45,46].

For the EFRP, the amplitude of the lambda response increased [47], and the latency was prolonged with time (Figure 9n,o). The reason for the increase in amplitude could be the effect of head sweat or the possibility that the participants tried to obtain more accurate information per saccade eye movement because the saccade frequency decreased. Latency was delayed when the VIMS symptoms became more severe. This may be due to a delay in visual information processing caused by the severity of VIMS [48]. More detailed studies focusing on the EFRP are required to clarify the relationship between the EFRP and VIMS.

As described above, the trends of changes in VIMS level and each feature were indicated, suggesting that the machine learning model approximately reflected the physiological changes. However, these relationships do not necessarily correspond to the importance of the features of the machine learning model (Table 4). This is because the lightGBM can also capture non-linear trends in features, and the feature importance could have a bias caused by variables with large cardinalities [49]. In addition, there is also a multicollinearity among the features. However, the physiological signals in this study are easily obtained with a low burden, so it is not important to reduce features. Although a minimum number of features is better from the viewpoint of calculation cost, it is desirable to acquire as many kinds of physiological signals as possible because of the risk of missing data due to the low burden. In addition, each acquired physiological index has a different origin. However, it is difficult to accurately classify VIMS utilizing many kinds of physiological indices, as in this study. Although the accuracy of the VIMS classification was higher than that in the previous studies, the accuracy varied among individuals (Table 4), which may be due to individual differences in physiological responses. However, it may also be possible to classify severe VIMS cases (Figure 9a).

## 7. Conclusions

In this study, we developed a low-burden physiological measurement system to detect VIMS, which is a workplace hazard in the remote-controlled excavator.

Using the developed system, physiological signals (ECG, Resp, EDA, EOG, and EEG) were measured in a simulated real work environment, and a machine learning model with LightGBM was constructed to estimate the VIMS level based on these signals. The constructed model was validated by LOOCV and scored 0.84 for the mean ROC-AUC and 0.71 for PR-AUC, indicating that the model is of a quality that is higher than that of models used in previous studies [3,8].

The model used all features related to physiological signals, with features considered particularly important being saccade frequency and SCR. Each feature was related to the VIMS level, and the relevant features differed from each other. Different relevance characteristics may have improved the quality of the model because the inability to evaluate the VIMS level with one feature could be compensated by another feature. Further studies are needed to determine where each feature compensates for the others in the model, such as by constructing a model using only the features related to each physiological signal and comparing them.

However, this study has some limitations. The proposed model did not perfectly predict the VIMS levels. As described in previous studies, there are limitations to predicting VIMS levels based on physiological signals alone. In the future, we plan to improve the model by utilizing not only physiological signals but also behavioral data (body movements and operation logs) to construct a multiclass classification model, thereby increasing the sensitivity to VIMS severity transitions.

## Figures and Tables

**Figure 1 sensors-24-06465-f001:**
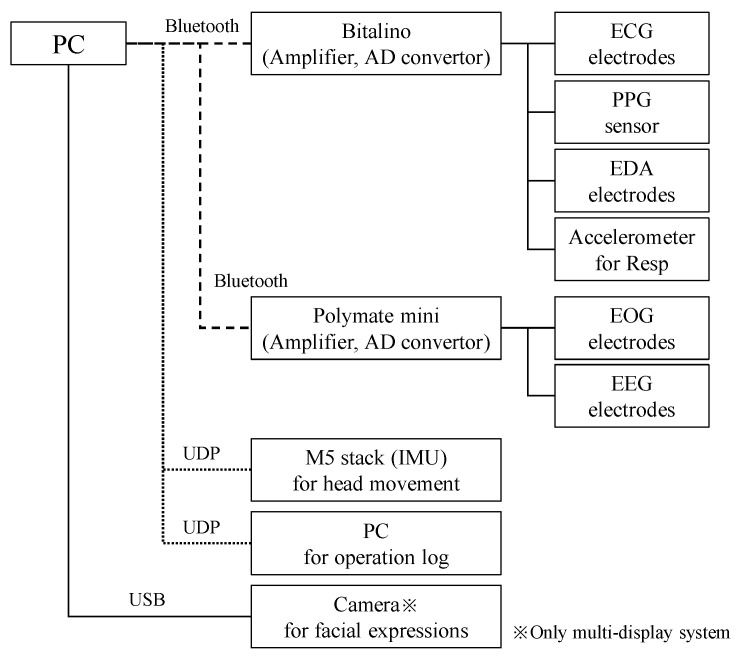
Block diagram of the prototype system.

**Figure 2 sensors-24-06465-f002:**
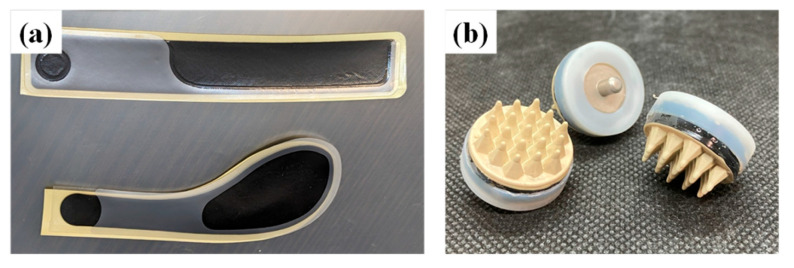
Dry electrodes. (**a**) Film electrodes for ECG, EDA, and EOG [10]. (**b**) Flexible dry electrodes for EEGs [11].

**Figure 3 sensors-24-06465-f003:**
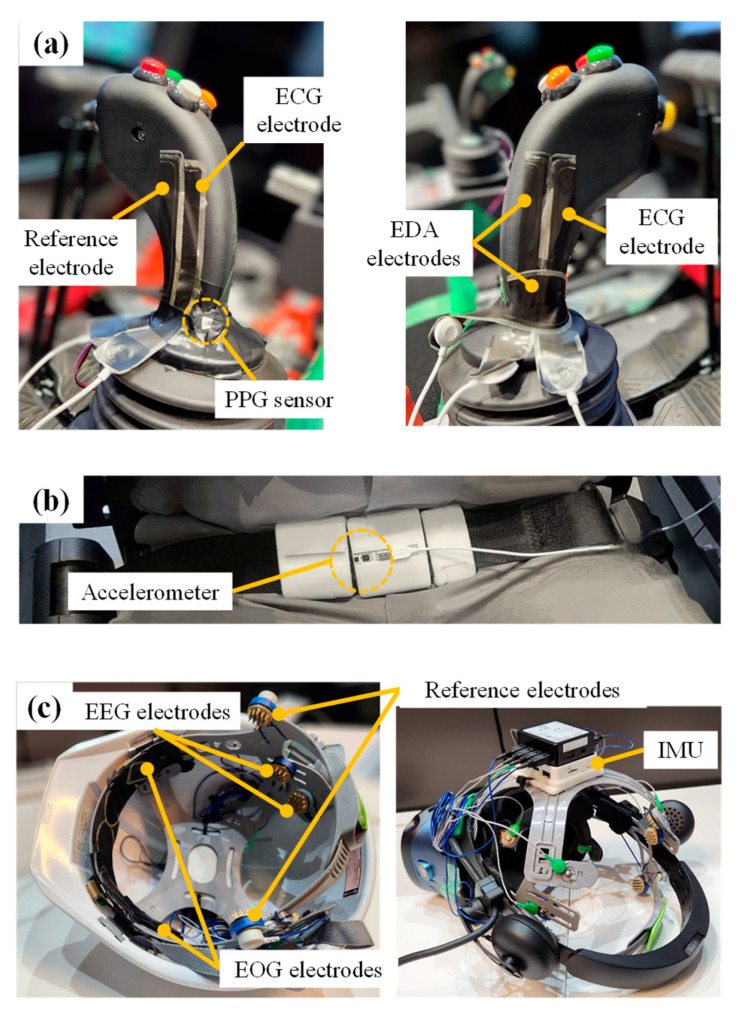
Low-burden physiological measurement system. (**a**) Levers with electrodes and sensor. (**b**) Seatbelt with an accelerometer. (**c**) Helmet and HMD with electrodes.

**Figure 4 sensors-24-06465-f004:**
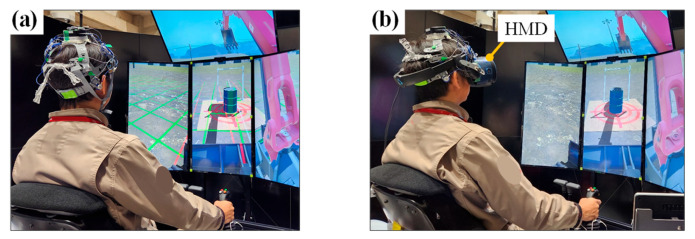
Experimental environment. (**a**) MD system: in this photo, the helmet comprises only the inner frame. (**b**) HMD system, with a multi-display only for experimenters.

**Figure 5 sensors-24-06465-f005:**
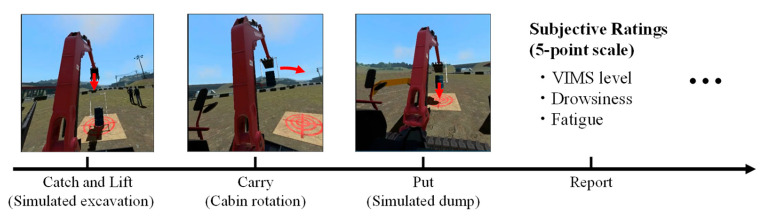
Experimental scenario, wherein excavation work with an excavator is simulated [14].

**Figure 6 sensors-24-06465-f006:**
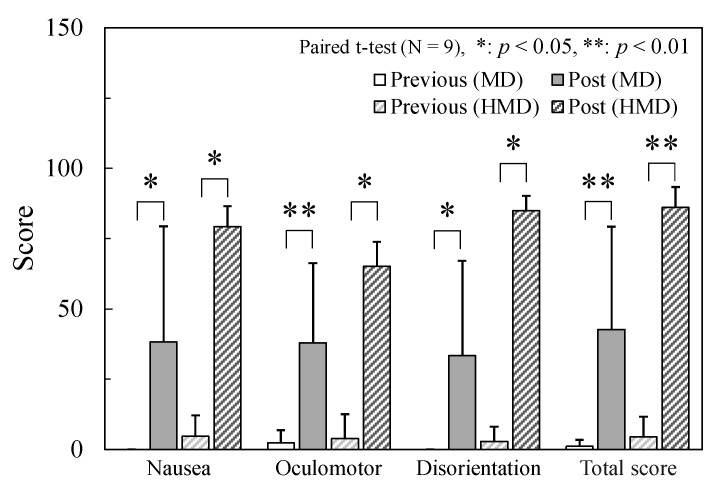
SSQ score pre- and post-experiment.

**Figure 7 sensors-24-06465-f007:**
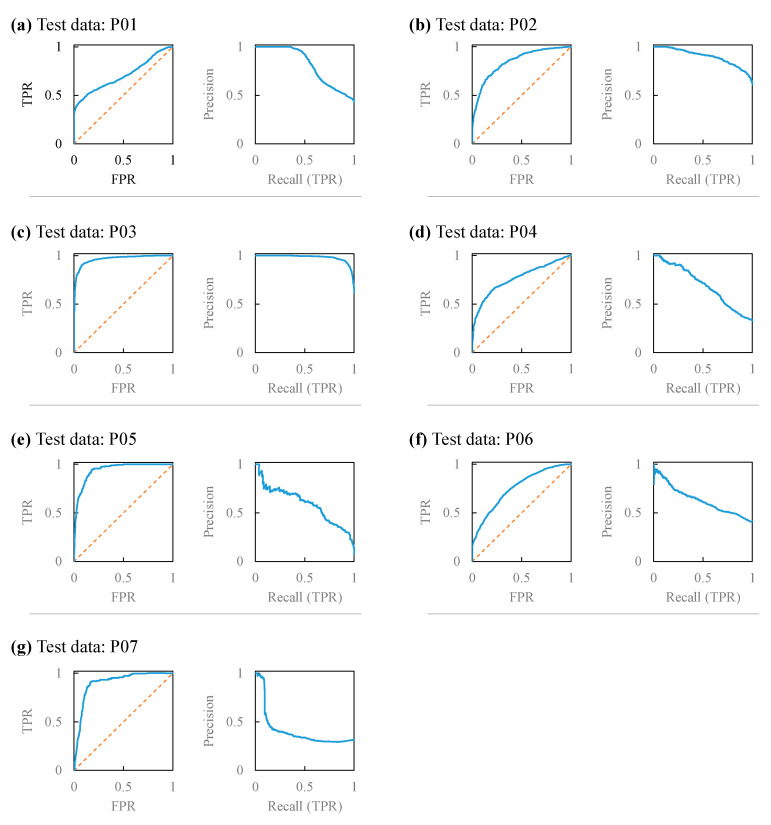
ROC curve and PR curve of (**a**) P01, (**b**) P02, (**c**) P03, (**d**) P04, (**e**) P05, (**f**) P06, and (**g**) P07.

**Figure 8 sensors-24-06465-f008:**
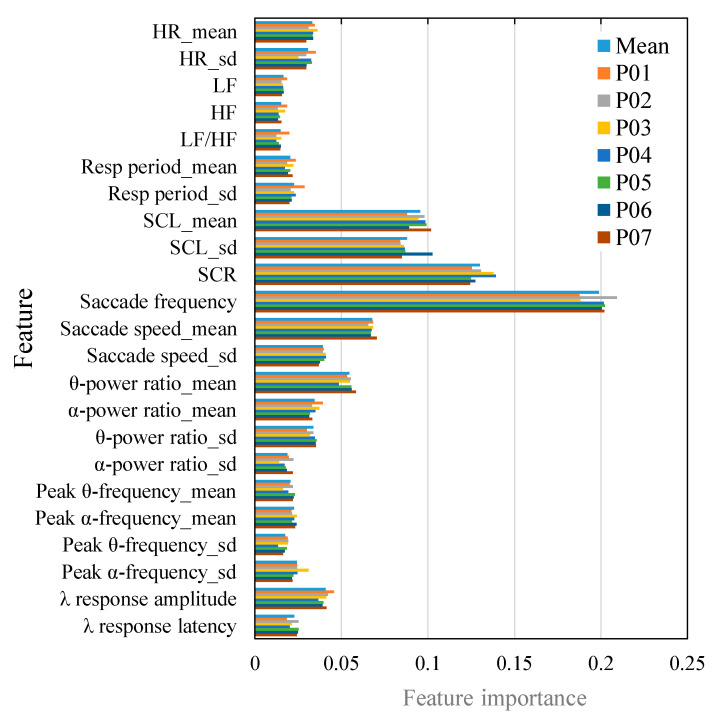
Feature the importance of each model and the mean value of all models.

**Figure 9 sensors-24-06465-f009:**
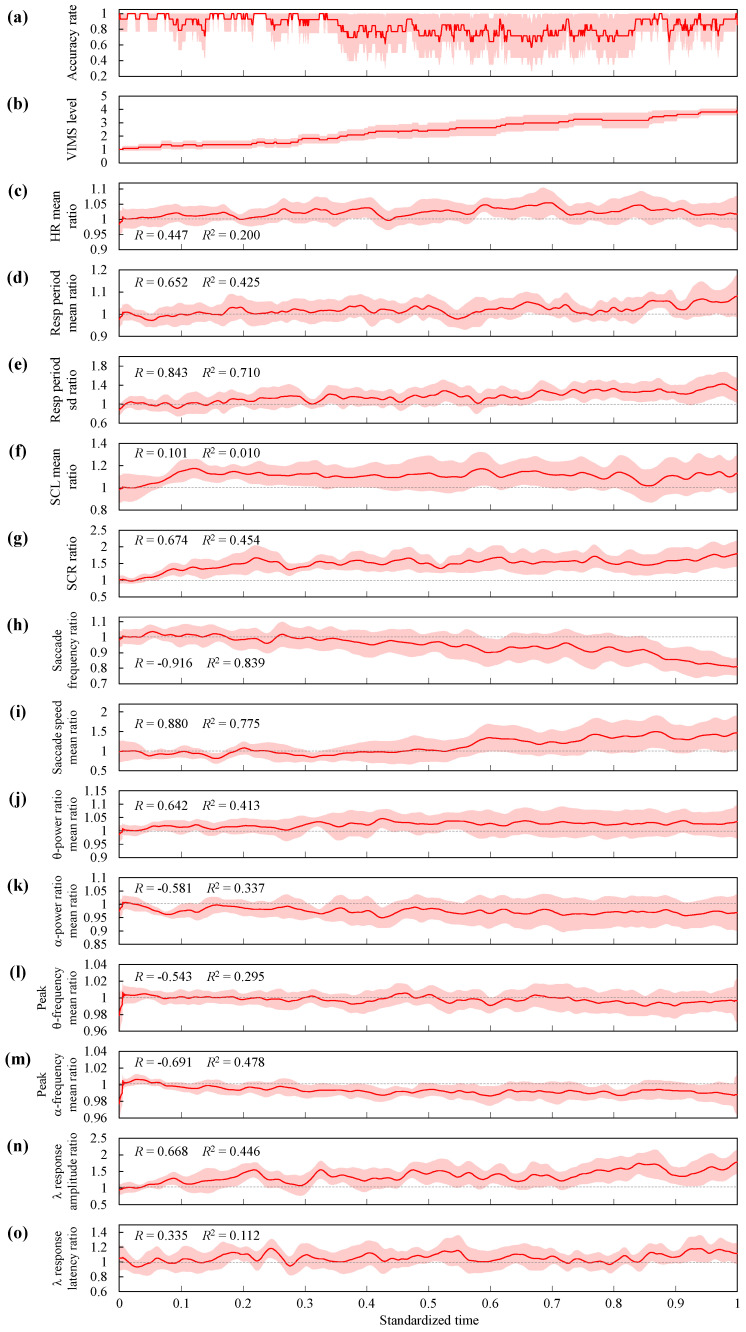
Change trends in (**a**) accuracy rate, (**b**) VIMS level, and (**c**–**o**) features. The red line indicates the mean, and the light red highlight indicates the 95% confidence interval. The gray dotted line in the graph of each feature indicates the baseline. *R* and *R*^2^ represent the correlation coefficients and the coefficients of determination with the VIMS level.

**Table 1 sensors-24-06465-t001:** Maximum value of the subjective evaluation of the VIMS levels in the subjective ratings for each experiment.

Participant	Maximum VIMS Level		Experimental Order	
	MD	HMD	Day1	Day2
P01	1	3	HMD	MD
P02	5	5	MD	HMD
P03	5	5	MD	HMD
P04	5	5	MD	HMD
P05	1	5	HMD	MD
P06	3	5	MD	HMD
P07	2	5	HMD	MD
P08	1	2	MD	HMD
P09	1	1	HMD	MD

**Table 2 sensors-24-06465-t002:** Number of time-series segments included in each dataset for the model construction.

Participants	Display Device	VIMS Level	
		1 or 2	3 or 4
P01	MD	940	-
	HMD	1342	1724
P02	MD	433	658
	HMD	271	464
P03	MD	738	1093
	HMD	313	616
P04	MD	1099	397
	HMD	450	959
P05	MD	1056	-
	HMD	1564	212
P06	MD	1185	736
	HMD	511	446
P07	MD	1777	-
	HMD	1848	212
P08	MD	617	-
	HMD	761	-
P09	MD	3545	-
	HMD	1544	-
Total		19,994	9659

**Table 3 sensors-24-06465-t003:** Accuracy, Kappa coefficient, and AUC scores of each test data.

Participant	Accuracy	*κ*-Coefficient	ROC-AUC	PR-AUC
P01	0.76	0.38	0.72	0.81
P02	0.82	0.66	0.84	0.90
P03	0.91	0.73	0.96	0.98
P04	0.80	0.39	0.76	0.69
P05	0.84	0.43	0.94	0.59
P06	0.67	0.32	0.76	0.63
P07	0.87	0.24	0.90	0.38
Mean	0.81	0.45	0.84	0.71

**Table 4 sensors-24-06465-t004:** List of feature importance, correlation coefficients, and coefficients of determination with the VIMS level.

Feature	Importance	*R*	*R* ^2^
HR_mean	0.033	0.447	0.200
HR_sd	0.031	−0.052	0.003
LF	0.017	0.154	0.024
HF	0.015	0.200	0.040
LF/HF	0.015	−0.125	0.016
Resp period_mean	0.020	0.652	0.425
Resp period_sd	0.023	0.843	0.710
SCL_mean	0.096	0.101	0.010
SCL_sd	0.088	0.587	0.345
SCR	0.130	0.674	0.454
Saccade frequency	0.199	−0.916	0.839
Saccade speed_mean	0.068	0.880	0.775
Saccade speed_sd	0.039	0.637	0.405
*θ*-power ratio_mean	0.055	0.642	0.413
*α*-power ratio_mean	0.034	−0.581	0.337
*θ*-power ratio_sd	0.034	−0.048	0.002
*α*-power ratio_sd	0.019	0.090	0.008
Peak *θ*-frequency_mean	0.021	−0.543	0.295
Peak *α*-frequency_mean	0.023	−0.691	0.478
Peak *θ*-frequency_sd	0.018	−0.022	0.000
Peak *α*-frequency_sd	0.024	−0.437	0.191
*λ* response amplitude	0.041	0.668	0.446
*λ* response latency	0.023	0.335	0.112

## Data Availability

The data and detailed parameters presented in this study are available upon request from the corresponding authors. The data are not publicly available due to confidentiality issues. Further, the datasets and program codes presented in this article are not readily available because they are part of an ongoing study.

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
