# Peer review of "Combined Method Comprising Low Burden Physiological Measurements with Dry Electrodes and Machine Learning for Classification of Visually Induced Motion Sickness in Remote-Controlled Excavator"

_sensors, 2024, doi:10.3390/s24196465_

Round 1

Reviewer 1 Report

Comments and Suggestions for Authors

This paper presents an interesting multi-sensor approach for classification of visually induced motion sickness. I find the topic very up-to-date and performed measurements valuable, but the overall representation and approach in signal analysis and classification must be improved.

Major comments:

1) Title is a bit vague. For example, how one defines what is low burden, especially with EEG measurements involved. Also, this paper deals with classification, I am not sure whether prediction should be used as a term. Please, revise the title.

2) Complete paragraph on visually induced motion sickness (lines 49-59) is without references. Authors should add relevant references.

3) More information of system evaluation (presented in [7]) on lines 65-66 is missing. Authors should add relevant conclusions, especially as this paper is not available.

4) I could not check, but if Figure 1 is from reference [7] than appropriate copyright permission should be obtained and stated.

5) Reference(s) for international 10-20 system is missing (line 74).

6) Is subjective score (line 100) introduced by Authors or has it been previously evaluated? Authors should add relevant information.

7) My major concern is related to data processing - how Authors determined time windows for data analysis (2 s, 30 s, and similar)?

8) Conversion of VIMS levels is not clear (lines 142-148). Maybe an illustration would help? Also, if this was repetition of levels range of time stamps, my question is how Authors assess errors that such repetition would provide? Would any other method for populating time stamps produce better results?

9) Procedure for determining hyperparameter tuning is missing. For example, I cannot tell: 1) which hyperparameters are tune, 2) what is the method for tuning hyperparameters, and 3) what are values of obtained hyperparameters.

10) Major concerns about machine learning application are: 1) why are just ROC and AUC used for evaluating classifier, 2) were data balanced and if not how disbalance was treated, and 3) was it justified to use feature importance, when importance can be affected by cross-correlation among features.

11) In the Discussion, Authors tried to explain machine learning results, but their method is not entirely correct as they commented relatively linear relationships. I would suggest to Authors to perform appropriate statistical analysis and then to compare results of statistical analysis and machine learning.

Minor comments:

1) LigthGBM stands for Light Gradient-Boosting Machine and Authors should explain this abbreviation at its first appearance in the paper

2) Table representation with list of extracted features, their expected, and obtained values would be useful to improve readibility and clarity.

3) The first paragraph from the Results section should be placed in the Methods section.

Author Response

Response to Reviewer 1 Comments

1. Summary

Thank you very much for taking the time to review this manuscript.

Especially the points made regarding the classification results were very useful. Our article was made better by your review.

Please find the detailed responses below and the corresponding revisions in the re-submitted files.

2. Questions for General Evaluation

Reviewer’s Evaluation

Does the introduction provide sufficient background and include all relevant references?

Can be improved

Is the research design appropriate?

Must be improved

Are the methods adequately described?

Must be improved

Are the results clearly presented?

Must be improved

Are the conclusions supported by the results?

Must be improved

3. Point-by-point response to Comments and Suggestions for Authors

Comments 1: Title is a bit vague. For example, how one defines what is low burden, especially with EEG measurements involved. Also, this paper deals with classification, I am not sure whether prediction should be used as a term. Please, revise the title.

Response 1: Thank you for pointing this out. We agree with this comment. Therefore, we have changed our article title. (lines 3-4)

Comments 2: Complete paragraph on visually induced motion sickness (lines 49-59) is without references. Authors should add relevant references.

Response 2: We have added 2 reference articles about VIMS in the remote-controlled excavator. (References 5,6)

Comments 3: More information of system evaluation (presented in [7]) on lines 65-66 is missing. Authors should add relevant conclusions, especially as this paper is not available.

Response 3: We have added an overview of the benefits of the system and the measured signals. (lines 71-74, and 79-84)

Comments 4: I could not check, but if Figure 1 is from reference [7] than appropriate copyright permission should be obtained and stated.

Response 4: Thank you for pointing this out. We do not use the same figures and pictures as in our previous paper.

Comments 5: Reference(s) for international 10-20 system is missing (line 74).

Response 5: We have added a reference article about the international 10-20 system. (References 13)

Comments 6: Is subjective score (line 100) introduced by Authors or has it been previously evaluated? Authors should add relevant information.

Response 6: Thank you for pointing this out. The subjective score was previously used in the co-author's other study. (References 15)

Comments 7: My major concern is related to data processing - how Authors determined time windows for data analysis (2 s, 30 s, and similar)?

Response 7: We added the explanation about the method of determining the size of the time-window. (lines 171-173, and References 23)

We wanted to generate samples for the dataset in as small a time segment as possible. The minimum time-window was a 30-second window. This is the minimum time window for the conversion from the ECG signals to the frequency component of heart rate.

Comments 8: Conversion of VIMS levels is not clear (lines 142-148). Maybe an illustration would help? Also, if this was repetition of levels range of time stamps, my question is how Authors assess errors that such repetition would provide? Would any other method for populating time stamps produce better results?

Response 8: We have revised the explanation of conversion of VIMS levels, and added a figure. (lines 175, and Figure 5)

We do not think that the errors caused by this conversion method have a significant impact on the results, but we recognize that it is one of the important issues, so we have added a description of other methods. (lines 319-327)

Comments 9: Procedure for determining hyperparameter tuning is missing. For example, I cannot tell: 1) which hyperparameters are tune, 2) what is the method for tuning hyperparameters, and 3) what are values of obtained hyperparameters.

Response 9: Thank you for pointing this out. We have added the method for tuning hyperparameters. (lines 239-243)

Comments 10: Major concerns about machine learning application are: 1) why are just ROC and AUC used for evaluating classifier, 2) were data balanced and if not how disbalance was treated, and 3) was it justified to use feature importance, when importance can be affected by cross-correlation among features.

Response 10: We have added about the evaluating classifier, and compared with the results of the previous study. (lines 246-253, 263-266)

In addition, we have added a discussion of the temporal trends of the classification results. (lines 303-305, 310-327, and Figure 9a)

The bias of the data was not balanced in this study. Although some of the data are biased, the percentage of biased data is not so large as a percentage of the total data, and therefore, the effect of bias is considered to be not significant. We understand that importance may be affected by cross-correlation among features. However, this has little effect on the classification results. The physiological measurement in this study is not the conventional method as in reference 3 but the low-burden physiological measurements for the purpose of practical application. So, there is a possibility of some signal acquisition failures. For this reason, we did not consider reducing the number of features in this study. (line 394-404)

Comments 11: In the Discussion, Authors tried to explain machine learning results, but their method is not entirely correct as they commented relatively linear relationships. I would suggest to Authors to perform appropriate statistical analysis and then to compare results of statistical analysis and machine learning.

Response 11: Thank you for pointing this out. Figure 9 was intended to confirm the trends of change. We have added an explanation of the validity of the comparison with the feature importance. (line 391-395)

Minor Comments 1: LigthGBM stands for Light Gradient-Boosting Machine and Authors should explain this abbreviation at its first appearance in the paper

Response M1: We have corrected this point. (line 156)

Minor comments 2: Table representation with list of extracted features, their expected, and obtained values would be useful to improve readibility and clarity.

Response M2: We have added Table 5.

Minor comments 3: The first paragraph from the Results section should be placed in the Methods section.

Response M3: We have revised the section. (line 245)

Reviewer 2 Report

Comments and Suggestions for Authors

Review of  “Combined Method Comprising Low Burden Physiological Measurements and Machine Learning-based Classification for Predicting Visually Induced Motion Sickness in Remote-controlled Excavator”

The study addresses the timely and important question of ways to reduce simulator sickness when teleoperating large machinery. The authors use the teleoperation of an excavation machine, which requires large turns during operation, such that the visual scenery moves substantially for the tele-operator. And the severity of VIMS (visually induces motion sickness) is known to be correlated with the extent of visual motion that is out of sync with the movement sensed by the vestibular system. Thus, they use an important case of tele-operation where reducing or preventing VIMS is of great importance. The attempt to relate physiological measures to subjective VIMS symptoms is worthwhile, and particularly the search for a contribution of eye-movements is novel and interesting. Also, the applied case of excavator teleoperation is a nice feature. It is superior to merely having subjects watch a nauseating film.

Major points:

Experimental Design: Researchers have attempted to measure VIMS by purely physiological measures for the longest time – and have failed to do so. Individual measures, such as galvanic skin conductance, are correlated with VIMS, but typically not sufficiently so to allow for good predictions. It is of course conceivable that an intricate combination of many physiological measures could in principle predict VIMS. And recently, some progress has been made by exploiting machine learning (see reference 3). Given the long and sobering tradition of finding physiological correlates of VIMS, it is a bit odd that the authors did not pick up where Keshavarz and colleagues left off. Given the title of the paper, one could have expected the exploration of thus far neglected physiological parameters with the established tools. The authors have nicely added eye-movements to the game, however, they have decided to throw away most of the differentiation with which the subjective experience of VIMS can be measured. It is not clear why the 20-point FMS has not been used. In particular, given that all subjects received both conditions (with and without HMD), relating the time course of VIMS to the physiological measures is of utmost importance. The SSQ, which the authors have used, could not be applied during the experiment, but rating scales can. But why have the authors chosen to use a coarse unvalidated 5-point rating scale as opposed to the validated fast motion sickness scale (FMS), which has a much better resolving power with its 20 gradations? This is very unfortunate and should be discussed as a shortcoming if it cannot be cured.

Methods description: Many details are missing. Was the order of HMD and MD conditions counterbalanced or did everyone do MD first and HMD second, as implied by Table 1? If counterbalanced, how many subjects saw each order?

What on earth is restraint time? Why did some subjects end up doing 1 hour and others 2 hours? What was the average exposure time for the two conditions?

Small N: Only 9 subjects produced good data. Given the large individual differences in VIMS, this is very small indeed. Of the 9 subjects, only 6 experienced VIMS to speak of. Thus, the study is seriously underpowered. I would suggest to run more subjects.

Results and Data Analysis: The modeling of the physiological parameters is strangely at odds with the subjective data. The former have been sampled finely, and the latter have been reduced to merely two levels. This categorization throws out even more of the information that was extremely coarse to begin with. Also, the time plots of VIMS and the physiological parameters (Figure 8) are very nice, but they should be supplemented with R2 values, so the reader can judge how much each measure contributes to the overall model.

Interpretation: The authors claim that their model is superior to that used by Keshavarz et al. as well as that used by Liu et al. Why do they think this is so? Is it the addition of eye-movements? And if so, why should horizontal saccade speed increase with VIMS? My guess would be that this feature is due to the particular task of picking up the oil barrel and turning around quickly. The faster subjects executed the task, the faster their saccades should be. Do you have data on movement execution times? They are probably correlated positively with VIMS and might further improve the model, once they are entered as a parameter. A more extensive discussion of your major modeling contribution is in place. As far as I can tell, the addition of eye-movements to the model is what makes the manuscript novel and thus a contribution that goes beyond the existing papers from 2022. This needs to be highlighted in your manuscript.

Any addition of parameters has an a priori chance to improve a model fit. Please comment on the possible removal of parameters. For a useful prediction model for VIMS to emerge, one should have to measure as few physiological parameters as one can get away with. After all, it is very tedious to measure EEG, eye-movements, HR, etc. all at once and in a synchronized fashion.

Minor points:

ICT is introduced as an acronym but never used, you can drop it. (information and communication technology (ICT)

The statement “The symptoms of VIMS are similar to those of motion sickness,” is empty, since VIMS is a subset of MS.

In sum, the attempt to put together a model that uses exclusively physiological measures to predict VIMS is time honored and very difficult indeed. It is commendable that the authors have taken on this task. They should point out the limitations of their study, and they should comment on which parameters are necessary and which may be dispensable for a good model.

Author Response

Response to Reviewer 2 Comments

1. Summary

Thank you very much for taking the time to review this manuscript.

Our article was made better by your review. Especially the points made regarding the eye-movements were very useful. Our article was made better by your review.

Please find the detailed responses below and the corresponding revisions in the re-submitted files.

2. Questions for General Evaluation

Reviewer’s Evaluation

Does the introduction provide sufficient background and include all relevant references?

Yes

Is the research design appropriate?

Can be improved

Are the methods adequately described?

Must be improved

Are the results clearly presented?

Must be improved

Are the conclusions supported by the results?

Must be improved

3. Point-by-point response to Comments and Suggestions for Authors

Comments 1: Experimental Design: Researchers have attempted to measure VIMS by purely physiological measures for the longest time – and have failed to do so. Individual measures, such as galvanic skin conductance, are correlated with VIMS, but typically not sufficiently so to allow for good predictions. It is of course conceivable that an intricate combination of many physiological measures could in principle predict VIMS. And recently, some progress has been made by exploiting machine learning (see reference 3). Given the long and sobering tradition of finding physiological correlates of VIMS, it is a bit odd that the authors did not pick up where Keshavarz and colleagues left off. Given the title of the paper, one could have expected the exploration of thus far neglected physiological parameters with the established tools. The authors have nicely added eye-movements to the game, however, they have decided to throw away most of the differentiation with which the subjective experience of VIMS can be measured. It is not clear why the 20-point FMS has not been used. In particular, given that all subjects received both conditions (with and without HMD), relating the time course of VIMS to the physiological measures is of utmost importance. The SSQ, which the authors have used, could not be applied during the experiment, but rating scales can. But why have the authors chosen to use a coarse unvalidated 5-point rating scale as opposed to the validated fast motion sickness scale (FMS), which has a much better resolving power with its 20 gradations? This is very unfortunate and should be discussed as a shortcoming if it cannot be cured.

Response 1: Thank you for pointing this out. We agree with this comment. Therefore, we have described the experimental method in more detail and added the reason why we did not use the FMS scale. (line 109, 114-121, and figure 5)

Comments 2: Methods description: Many details are missing. Was the order of HMD and MD conditions counterbalanced or did everyone do MD first and HMD second, as implied by Table 1? If counterbalanced, how many subjects saw each order?

What on earth is restraint time? Why did some subjects end up doing 1 hour and others 2 hours? What was the average exposure time for the two conditions?

Response 2: We have described the experimental method in more detail. (line 104, 123-125, and Table 1)

The difference in experimental time was due to a malfunction of the large-scale experimental equipment.

Comments 3: Small N: Only 9 subjects produced good data. Given the large individual differences in VIMS, this is very small indeed. Of the 9 subjects, only 6 experienced VIMS to speak of. Thus, the study is seriously underpowered. I would suggest to run more subjects.

Response 3: We agree with this comment. But in this experiment, it was necessary to request the participation of operators with specialized skills, rather than the general public.

This was difficult because, as the background indicates, there is a shortage of workers, and many of them are working regardless of weekdays and weekends in the construction industry. Unfortunately, the number of them registered in the database of experimental participants is also small. In addition, the large scale of the experiments also makes it difficult to occupy the experimental equipment.

However, we think that the trends in the relationship between VIMS and physiological responses can be observed in this study. And we will use the results of this study as a bargaining chip to further recruit participants and use the experimental equipment.

Comments 4: Results and Data Analysis: The modeling of the physiological parameters is strangely at odds with the subjective data. The former have been sampled finely, and the latter have been reduced to merely two levels. This categorization throws out even more of the information that was extremely coarse to begin with. Also, the time plots of VIMS and the physiological parameters (Figure 8) are very nice, but they should be supplemented with R2 values, so the reader can judge how much each measure contributes to the overall model.

Response 4: We have added the reason for the binary classification and the list of the result including R2 values. (line 120, and Table 5)

Comments 5: Interpretation: The authors claim that their model is superior to that used by Keshavarz et al. as well as that used by Liu et al. Why do they think this is so? Is it the addition of eye-movements? And if so, why should horizontal saccade speed increase with VIMS? My guess would be that this feature is due to the particular task of picking up the oil barrel and turning around quickly. The faster subjects executed the task, the faster their saccades should be. Do you have data on movement execution times? They are probably correlated positively with VIMS and might further improve the model, once they are entered as a parameter. A more extensive discussion of your major modeling contribution is in place. As far as I can tell, the addition of eye-movements to the model is what makes the manuscript novel and thus a contribution that goes beyond the existing papers from 2022. This needs to be highlighted in your manuscript.

Response 5: Thank you for pointing this out. The reason for good classification performance of our models is thought to be the utilization of many different physiological indices (autonomic nervous system indices, eye movements, and EEG) for the features.

Keshavarz et al. and Liu et al. used only either autonomic nervous system indices or EEG.

We have added our discussion about the eye movements. (line 365-375)

As shown in the graph below, the cabin rotation speed during the operation did not increase much. The coefficient of determination: R2 = 0.059 (with the VIMS level).

Of course, further data measurement is needed in order to better elucidate these issues.

Comments 6: Any addition of parameters has an a priori chance to improve a model fit. Please comment on the possible removal of parameters. For a useful prediction model for VIMS to emerge, one should have to measure as few physiological parameters as one can get away with. After all, it is very tedious to measure EEG, eye-movements, HR, etc. all at once and in a synchronized fashion.

Response 6: We have added thoughts about feature reduction. (line 395-399)

Certainly, it is difficult to measure multiple physiological signals synchronously with the conventional method. However, with our system, these signals can be easily measured during normal excavator operation. (line 79-81)

Minor comments 1: ICT is introduced as an acronym but never used, you can drop it. information and communication technology (ICT)

Response M1: We have modified it. (line 36-37)

Minor comments 2: The statement “The symptoms of VIMS are similar to those of motion sickness,” is empty, since VIMS is a subset of MS.

Response M2: We have modified it. (line 52-53)

Minor comments 3: In sum, the attempt to put together a model that uses exclusively physiological measures to predict VIMS is time honored and very difficult indeed. It is commendable that the authors have taken on this task. They should point out the limitations of their study, and they should comment on which parameters are necessary and which may be dispensable for a good model.

Response M3: We have added thoughts about feature reduction. (line 395-399)

Round 2

Reviewer 1 Report

Comments and Suggestions for Authors

I am very satisfied with answers and changes. I have two minor comments:

Authors did not properly answered to Comment 10. Cross-correlation creates a bias when feature importance measures are used. I think that at least Authors should address this issue. Here's a sample reference, but there are many more (Authors can cite reference at their choice):

Adler, A. I., & Painsky, A. (2022). Feature importance in gradient boosting trees with cross-validation feature selection. Entropy, 24(5), 687.   Also, for Comment 11, Authors missed the point. I wanted to say that machine learning can detect non-linear trends and/or complex inter-relations among features beyond linear relationships.

Author Response

Thank you for your important remarks and for recommending an informative article that will help us to better understand the issue. We have added a note about the feature importance and referred to the article you mentioned. (lines 393-395, reference 49)

Reviewer 2 Report

Comments and Suggestions for Authors

Review of revised manuscript  “Combined Method Comprising Low Burden
Physiological Measurements with Dry Electrodes and Machine Learning for
Classification of Visually Induced Motion Sickness in Remote-controlled
Excavator”

The authors have nicely responded to my concerns. In particular, the
addition of Figure 5 is very helpful.

Merely a statement as to how practical the full package of physiological
measures might be given that there is still a lot of variance in the
data could be helpful.

As a comment, I think for future studies, the authors should consider
using a more fine-grained scale to measure VIMS, like the FMS. It is not
more cumbersome or interfering with the task as is a 5-point scale, and
it would provide better quantitative information. 

Author Response

Thank you for your comment. In this study, even if there are multi-point scale, the model uses binary values, so we adopted a 5-point scale, giving priority to the viewpoint of reducing the burden and hesitation of the participants in the experiment. However, since your opinion is important, we would like to conduct a basic experiment in the future to compare FMS and 5-point scale, and to examine whether the model can be improved when FMS is used.